

# High-throughput sequencing reveals rhizosphere fungal community composition and diversity at different growth stages of *Populus euphratica* in the lower reaches of the Tarim River

Yuanyuan Li, Hanli Dang, Xinhua Lv, Zhongke Wang, Xiaozhen Pu and Li Zhuang

Shihezi University, Shihezi, Xinjiang Uygur Autonomous Region, China

## ABSTRACT

**Background**. *Populus euphratica* is one of the most ancient and primitive tree species of *Populus spp* and plays an important role in maintaining the ecological balance in desert areas. To decipher the diversity, community structure, and relationship between rhizosphere fungi and environmental factors at different growth stages of *P. euphratica* demands an in-depth investigation.

**Methods**. In this study, *P. euphratica* at different growth stages (young, medium, overripe, and decline periods) was selected as the research object, based on the determination of the physicochemical properties of its rhizosphere soil, the fungal community structure and diversity of *P. euphratica* and their correlation with soil physicochemical properties were comprehensively analyzed through high-throughput sequencing technology (internal transcribed spacer (ITS)) and bioinformatics analysis methods.

**Results**. According to the analysis of OTU annotation results, the rhizosphere soil fungal communities identified in *Populus euphratica* were categorized into 10 phyla, 36 classes, 77 orders, 165 families, 275 genera and 353 species. The alpha diversity analysis showed that there was no obvious change between the different growth stages, while beta diversity analysis showed that there were significantly differences in the composition of rhizosphere soil fungal communities between mature and overripe trees ($R^2 = 0.31$, $P = 0.001$), mature and deadwood ($R^2 = 0.28$, $P = 0.001$). Ascomycota and Basidiomycota were dominant phyla in the rhizosphere fungal community and the dominant genera were *Geopora, Chondrostereum* and *unidentified_Sordariales_sp*. The relative abundance of the top ten fungi at each classification level differed greatly in different stages. Canonical correspondence analysis (CCA) and Spearman's correlation analysis showed that conductivity (EC) was the main soil factor affecting the composition of *Populus euphratica* rhizosphere soil fungal community ($P < 0.01$), followed by total dissolvable salts (TDS) and available potassium (AK) ($P < 0.05$).

**Conclusions**. Our data revealed that the rhizosphere fungal communities at the different growth stages of *P. euphratica* have differences, conductivity (EC) was the key factor driving rhizosphere fungi diversity and community structure, followed by total dissolvable salts (TDS) and available potassium (AK).

Corresponding authors
Xiaozhen Pu, 2609240380@qq.com
Li Zhuang, 3179673644@qq.com

## INTRODUCTION

Soil microorganism is one of the key components of desert soil microecosystem, which plays an important role in soil nutrient cycle and vegetation nutrient supply and recovery (*Neilson et al., 2012*; *Bull & Asenjo, 2013*). Rhizosphere soil fungi are an important part of soil-plant ecosystem, and 'their community structure closely affects the growth and development of plants. The diversity and ecological function of rhizosphere soil fungi have become a hot topic in the field of soil microecology at home and abroad (*Wu, Lin & Lin, 2014*). Previous studies on soil fungal communities are mainly based on the traditional plate culture method, which has proved the dependence of plants on fungal communities (*Costa et al., 2006*). However, the number of soil fungi obtained by this method is very small, and many fungi cannot be isolated and cultured directly (*Wang et al., 2021a*; *Wang et al., 2021b*), so that it cannot fully reflect the composition of the community structure.

With the development of science and technology, high-throughput sequencing technology has gradually become the main method to study soil microorganisms (*Liu et al., 2015*; *Luo et al., 2020*). In the current research, it was found that there is a close correlation between rhizosphere microorganisms and plants, and their interaction mechanism was complicated (*Morgan, Bending & White, 2005*). The composition of rhizosphere microbial community will be affected by vegetation type (*Sinha et al., 2008*), soil type (*Lu et al., 2011*; *Acharya et al., 2021*), human factors and other factors (*Li et al., 2015*; *Huang et al., 2021*; *Yin, Li & Du, 2021*). For the same plant, different planting methods (*Durrer et al., 2021*), different development stages, even genetic background and other factors will lead to the change of the rhizosphere microbial community (*Marschner et al., 2001*; *Dang et al., 2020*).

*Populus euphratica* is one of the most ancient and primitive tree species of *Populus spp*. It is a unique desert forest tree species, having the characteristics of drought resistance, saline alkali resistance, heat resistance, wind and sand resistance, and plays an important role in maintaining the ecological balance in desert areas (*Nekoa et al., 2018*). China has the largest distribution range and the largest number of *P. euphratica* species in the world. More than 90% of *P. euphratica* forests in China are concentrated in Xinjiang region, and mainly in the lower reaches of Tarim River and many downsteam in the southern edge of Tarim basin (*Wang, 1996*). However, some researchers indicated that *P. euphratica* population regeneration in the lower reaches of Tarim River showed a decline type, the proportion of young plants in the population decreased significantly or even lacked, population was mostly over mature forest plants and the overall performance of the decline trend (*Zhou et al., 2018*). Currently, most studies of *P. euphratica* were focus on heteromorphic leaves (*Li et al., 2020a*; *Li et al., 2020b*), photosynthetic physiological characteristics (*Wang et al., 2014*), water use efficiency (*Zhou et al., 2019*), population structure (*Miao et al., 2020*), *etc*, while few studies on the relationship between plant and rhizosphere soil fungal community composition and diversity. Therefore, the goal of our work was to (1)

**Table 1  Morphological characteristics of *P. euphratica* at different growth stages.**

| Group | $\phi$/cm | H/m | P/m $\times$ m |
|---|---|---|---|
| A | 3.50~5.41 | 2.53~3.25 | 2.02 $\times$ 1.93~2.62 $\times$ 3.13 |
| B | 8.12~9.62 | 4.50~5.50 | 2.82 $\times$ 3.20~3.50 $\times$ 3.98 |
| C | 59.23~67.88 | 12.30~16.50 | 6.21 $\times$ 6.09~5.86 $\times$ 7.40 |
| D | | / | |

Notes.

$\phi$, diameter at breast height (DBH); H, height of tree; P, crown; A, sapling; B, mature; C, overripe; D, deadwood.

analyze the composition and diversity of rhizosphere soil fungi of *P. euphratica* at different growth stages in the lower reaches of the Tarim River; (2) explore the dominant fungi in the *P. euphratica* rhizosphere at different stages, and the change of rhizosphere soil physical and chemical properties; (3) elucidate the correlation between fungal community composition and environmental factors. This study will provide scientific basis for the study of rhizosphere microorganisms and population rejuvenation of *P. euphratica* and the interaction between plants and microorganisms in arid areas.

## MATERIALS & METHODS

### Study sites and sampling

The study area was located in the natural *P. euphratica* forest in the lower reaches of Tarim River basin, Xinjiang province (with the geographical coordinates of 40°28′~40°55′N, 87°51′~87°75′E), China. This area belonged to a typical continental extreme arid climate, with the annual average precipitation was less than 50 mm, the evaporation was about 2,960 mm, the annual total solar radiation was 5,692~6,360 kJ m$^{-2}$, and the annual average temperature was 10.5 ~11.4 °C (*Yang & He, 2000*), the ecological environment was extremely bad.

In mid-September 2020, soil samples were collected from selected natural *P. euphratica* forests. According to the classification standard in *P. euphratica* forest written by *Wang, Chen & Li (1995)*. Four growth stages are selected, which including sapling (A), mature (B), overripe (C), and deadwood (D). The diameter at breast height (DBH) of sapling was about four cm, mature wood was 4~10 cm, and overripe wood was 30~70 cm. Three trees with similar growth and no diseases and insect pests were selected for each stage to measure morphological characteristics (Table 1) and collect rhizosphere soil samples (60 cm). At the same time, the bare land without vegetation cover was selected as blank control (CK), in the area and soil samples were collected (same depth), set up three sampling points to take mixed soil samples, and obtain three groups of parallel samples at each place.

The sampling method of rhizosphere soil microorganisms was to dig the soil profile 0.5 m away from the primary root, and start from the fine root (sample's depth was 60 cm), the soil adhered to the root segment after shaking the fine root was the rhizosphere soil. The collected soil was divided into three parts, one part was placed in a five mL sterile centrifuge tube and stored in a liquid nitrogen tank for the determination of rhizosphere fungal community; other part soil samples were put into sealed bags and brought back to the laboratory, after natural air-drying, they were screened (two mm) for the determination
of soil physical and chemical properties; the last part of the soil samples were weighed in aluminum boxes, and drying in an oven (72°/48 h) at the laboratory used for soil moisture measurement.

## Soil physicochemical properties

Soil properties were assessed as described in prior studies (*Bao, 2008*), organic matter content (OM) was assessed using the $KCr_2O_7$ method, total nitrogen(TN) was assessed using the $HClO_4$-$H_2SO_4$ digestion method, total phosphorus (TP) was assessed using a Mo-Sb colorimetric method, total potassium (TK) was measured via atomic absorption spec-trometry, nitrate nitrogen(SNN), ammonium nitrogen(SAN) was assessed via a 0.01 M calcium chloride extraction method using a BRAN+LUEBBE flow analyzer, available phosphorus(AP) wasdetermined by molybdenum antimony anti Colorimetry (sodium bicarbonate extraction), available potassium (AK) was determined by atomic absorption spectrometry (ammonium acetate extraction), pH (as measured with a Mettler Tolido FiveEasy Plus pH meter), total dissolvable salts (TDS) (as assessed via atomic absorption spec-trometry and titration), and conductivity (EC) (measured by Hanna H1 2315 conductivity meter). In addition, the above determination of soil physical and chemical properties was repeated three times.

## Soil fungi DNA extraction, PCR amplification and sequencing

Most of the methods adopted here are previously described in *Hu, Yesilonis & Szlavecz (2021)*. Briefly, soil genomic DNA was extracted from rhizosphere soil samples of *P. euphratica* by cetyltrimethylammonium bromide (CTAB) (*Hu, Yang & You, 2010*), after that, the purity and concentration of DNA were detected by 2% agarose gel electrophoresis. A proper amount of DNA sample was taken into a centrifuge tube and diluted with sterile water to 1 ng $\mu L^{-1}$. Using diluted genomic DNA as a template, ITS1 primers ITS5-1737F (5′-GGAAGTAAAAGTCGTAACAAGG-3′) (*Bellemain et al., 2010*) and ITS2-2043R (5′-GCTGCGTTCTTCATCGATGC-3′), Phusion® high-fidelity PCR Master Mix with GC Buffer and efficient high-fidelity enzyme from Biolabs, New England, were selected for PCR (*Walters et al., 2016*). The PCR product was detected by electrophoresis using 2% agarose gel, and recovered using the gel recovery kit provided by Qiagen company, TruSeq® DNA PCR-free Sample Preparation Kit (Illumina, USA) was used to construct the library, which was quantitated by Qubit and Q-PCR. Lastly, the constructed library was sequenced and computerized on Illumina HiSeq2500 platform of Beijing Compson Biotechnology Co., Ltd.

## Sequence processing and analysis

FLASH (V1.2.7, http://ccb.jhu.edu/software/FLASH/) (*Magoč & Salzberg, 2011*) was used to splice the offline data obtained by sequencing to get Raw tags data, through the Qiime (V1.9.1, http://qiime.org/scripts/split_libraries_fastq.html) (*Gregory et al., 2010*) for data quality control, after strict filtering (*Bull & Asenjo, 2013*), clean tags are obtained, then we do chimera filtering (https://github.com/torognes/vsearch/) (*Haas et al., 2011*), and ultimately get can be used for further analysis of Effective tags.

Uparse (v7.0.1001, http://www.drive5.com/uparse/) (*Edgar, 2013*) was used to cluster sequences into Operational taxa (OTUs) with 97% sequence similarity, annotate the OTUs sequences, analyze the species annotation with the BLAST method (http://qiime.org/scripts/assign_taxonomy.html) (*Altschul, 1990*) in Qiime software and the UNITE (v8.2) database (https://unite.ut.ee/) (*Kõljalg et al., 2013*), and count the community composition of each sample at each classification level. The MUSCLE (v3.8.31) (http://www.drive5.com/muscle/) (*Edgar, 2004*) was used for fast multi-sequence alignments to get the phylogenetic relationships of all OTUs sequences, the data of each sample was homogenized, and the data with the least amount of data in the sample was used as the standard for homogenization. The subsequent alpha diversity and beta diversity were analyzed based on the homogenized data.

Qiime (Version 1.9.1) was used to calculate fungal diversity index of soil samples, including microbial richness (Chao1 index and ACE index) and microbial diversity (Shannon index and Simpson index) (*Wang et al., 2019*).

R (Version 2.15.3) was used to draw the dilution curve, Venn diagram and Principal coordinates analysis (PCoA) diagram. The Analysis of similarities (Adonis)in "vegan" R package was used to examine differences between groups. LEfSe software was used for linear discriminant analysis and effect size analysis with the default filtering value of LDA score set at 4. Canonical correlation analysis (CCA) was used to test the relationship among environmental factors, samples and microbes. The CCA was estimated using the "vegan" package in R (v3.6.1). Correlations between soil physicochemical properties and fungal community composition were assessed via Spearman's correlation analyses.

Statistical analysis (including one-way analysis of variance (ANOVA) and Spearman's correlation analysis) were carried out with SPSS 22.0 (IBM Inc., Armonk, USA).

## RESULTS

### Differences in physical and chemical properties of rhizosphere soil

The physical and chemical properties of soil samples were measured (Table 2). The content of SWC in A sample was significantly higher than B and C samples ($P < 0.05$), however, there was no significant difference in the contents of TP, SNN and SAN in *P. euphratica* rhizosphere soil at different growth stages ($P > 0.05$). The content of OM, TN, TK, AK, AP, pH, EC and TDS in D sample were higher than A, B and C samples, among which the contents of TK, AK, pH, EC and TDS were significantly different ($P < 0.05$). In addition, there was no significant difference between B and C samples except the content of AK ($P > 0.05$).

### Sequencing data and OTU clustering

By sequencing the ITS fragments of soil fungi, 947,049 effective sequences were obtained from soil samples, and 960 OTUs were obtained by clustering with 97% sequence similarity. The dilution curves of all soil samples tended to be flat, indicating that the sequencing depth had basically covered all fungal groups in the samples, which could reflect the real situation of soil fungal community in the rhizosphere of *P. euphratica* (Fig. 1).
**Table 2 Physicochemical properties of rhizosphere soil of *P. euphratica* at different growth stages.**

|  | A | B | C | D | CK |
|---|---|---|---|---|---|
| SWC (%) | 6.34 ± 0.65a | 3.41 ± 1.11bc | 3.17 ± 1.05bc | 5.02 ± 0.24ab | 1.61 ± 0.09c |
| OM (g kg⁻¹) | 7.06 ± 1.66ab | 7.17 ± 2.13ab | 5.66 ± 1.11b | 11.04 ± 1.44a | 5.16 ± 0.78b |
| TN (g kg⁻¹) | 0.42 ± 0.07ab | 0.51 ± 0.03ab | 0.37 ± 0.04b | 0.54 ± 0.02a | 0.41 ± 0.02ab |
| TP (g kg⁻¹) | 0.58 ± 0.01a | 0.61 ± 0.03a | 0.61 ± 0.05a | 0.64 ± 0.01a | 0.65 ± 0.03a |
| TK (g kg⁻¹) | 18.25 ± 0.54b | 17.54 ± 0.18bc | 17.93 ± 0.23b | 19.71 ± 0.15a | 16.85 ± 0.23c |
| AK (g kg⁻¹) | 0.42 ± 0.03c | 0.34 ± 0.12c | 0.93 ± 0.21b | 3.40 ± 0.24a | 0.39 ± 0.08c |
| AP (mg kg⁻¹) | 3.06 ± 1.83ab | 2.01 ± 1.45b | 1.86 ± 1.09b | 6.03 ± 0.93a | 2.31 ± 0.71ab |
| SNN (mg kg⁻¹) | 4.86 ± 1.04ab | 3.16 ± 2.19b | 2.84 ± 0.87b | 3.97 ± 0.43ab | 8.14 ± 2.35a |
| SAN (mg kg⁻¹) | 2.49 ± 0.04a | 2.94 ± 0.41a | 2.46 ± 0.69a | 2.11 ± 0.31a | 2.72 ± 0.21a |
| pH (1:5) | 7.78 ± 0.05c | 8.43 ± 0.13b | 8.61 ± 0.15b | 9.49 ± 0.19a | 7.89 ± 0.19c |
| EC (ms cm⁻¹) | 2.78 ± 0.15b | 2.11 ± 1.46b | 2.74 ± 1.08b | 7.96 ± 1.14a | 5.34 ± 0.78ab |
| TDS (g kg⁻¹) | 9.78 ± 0.42b | 7.81 ± 5.34b | 8.65 ± 3.88b | 30.01 ± 4.59a | 18.84 ± 2.98ab |

**Notes.**

A, sapling; B, mature; C, overripe; D, deadwood; CK, bare soil; SWC, soil water content; OM, organic matter; TN, total nitrogen; TP, total phosphorus; TK, total potassium; AK, available potassium; AP, available phosphorus; SNN, nitrate nitrogen; SAN, ammonium nitrogen; pH, hydrogen ion concentration; EC, electrical conductivity; TDS, total salt.
Values in the table are mean ± standard deviation, different letters in the same line indicate significant differences ($p < 0.05$).

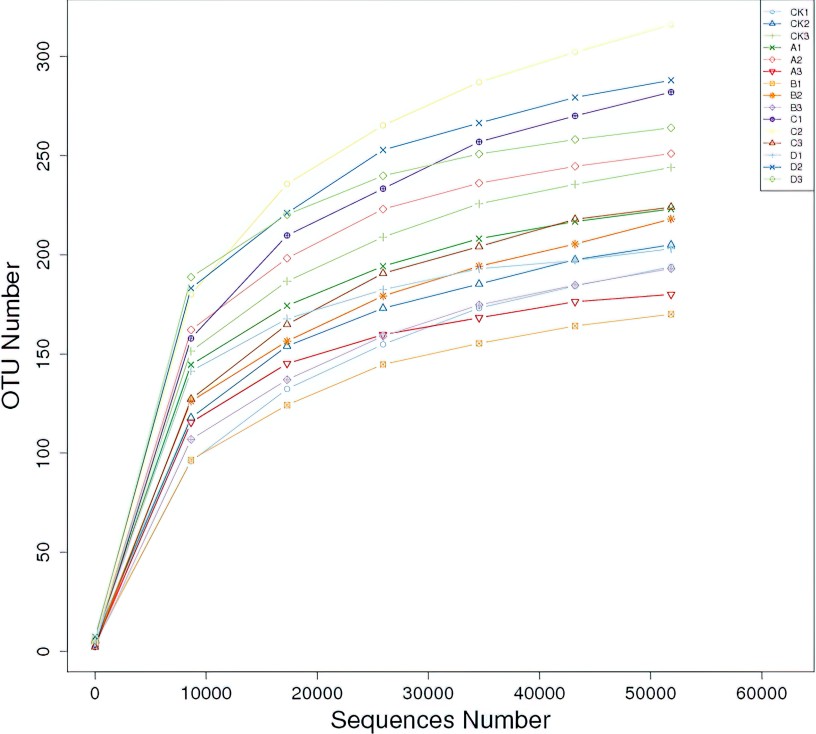

**Figure 1   Rarefaction curves of fungal community composition in 15 samples.** The rarefaction curves different colors represent different samples (CK, A, B, C and D: bare soil, sapling, mature, overripe and deadwood, respectively; the second number representing the replicate number).

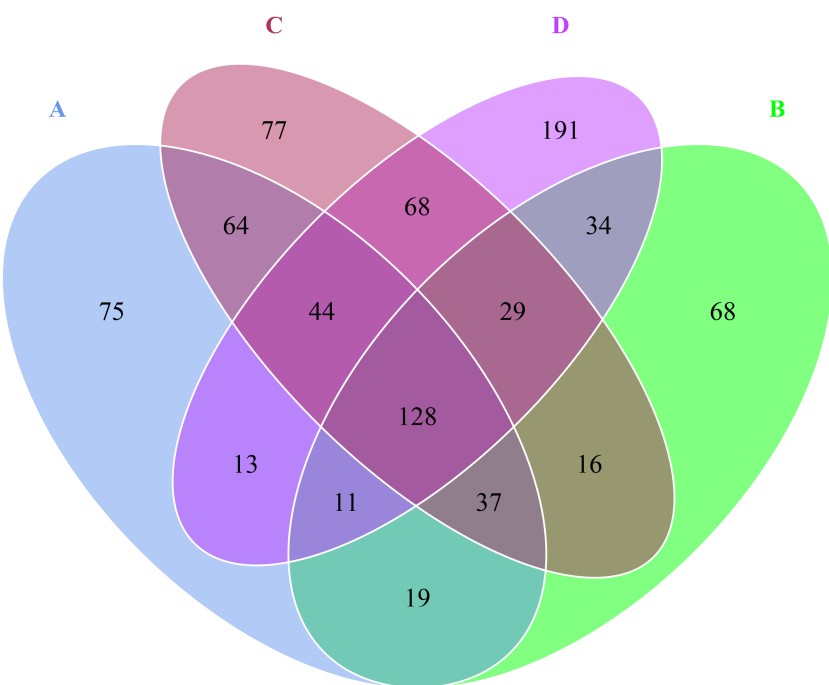

**Figure 2 Venn diagram at OTU level in the rhizosphere of *P. euphratica*.** Each petal corresponds to a sample group, with the shared overlapping region representing OTUs common to all samples, and the numbers on individual petals representing the number of OTUs unique to a given sample group. (A, B, C and D: bare soil, sapling, mature, overripe and deadwood).

Venn diagram revealed that there were 391 OTUs for A sample, 342 OTUs for B sample, 463 OTUs for C sample, 518 OTUs for D sample, 128 OTUs were shared by A, B, C and D samples (Fig. 2). Besides, the numbers of unique OTUs to each sample were as follows: 75 for A sample, 68 for B sample, 77 for C sample and 191 for D sample, respectively, accounting for 19.18%, 19.88%, 16.63% and 36.87% of the all OTUs.

## Differences in fungal diversity

The alpha diversity index of soil fungal community in rhizosphere of *P. euphratica* at different growth stages was different (Table 3). As shown in Table 3, the Coverage index of each sample was close to 100%, which proved the integrity of the detected samples sequence, indicating that the sequencing results at this level could reflect the true situation of fungal community composition in the measured samples. Shannon index was consistent with Simpson index, the value was the highest in D sample, Chao1 index and ACE index were the highest in B sample. However, there was no significant difference between different growth stages.

Both PCoA (Fig. 3) and Adonis (Table 4) all revealed that there were significant differences in soil fungal community composition between B and C samples ($R^2 = 0.31$ $P = 0.001$), B and D samples ($R^2 = 0.28$ $P = 0.001$). There was no significant difference between A and the other samples.
**Table 3 Diversity indices for each sample.**

| Group | Shannon | Simpson | Chao1 | ACE | Coverage |
|-------|---------|---------|-------|-----|----------|
| CK | 3.234 ± 0.388 | 0.791 ± 0.045 | 232.519 ± 24.107 | 236.779 ± 23.966 | 0.999 ± 0 |
| A | 2.219 ± 0.419 | 0.543 ± 0.119 | 277.532 ± 77.002 | 240.901 ± 34.103 | 0.999 ± 0 |
| B | 1.887 ± 0.470 | 0.396 ± 0.118 | 309.919 ± 29.289 | 318.324 ± 32.015 | 0.999 ± 0 |
| C | 3.136 ± 0.767 | 0.706 ± 0.120 | 238.784 ± 14.176 | 244.780 ± 13.354 | 0.999 ± 0 |
| D | 3.989 ± 0.562 | 0.813 ± 0.094 | 272.700 ± 26.691 | 272.206 ± 27.908 | 0.999 ± 0 |
| $p$ | 0.114 | 0.0761 | 0.672 | 0.258 | |

**Notes.**
Values in the table are mean ± standard deviation.

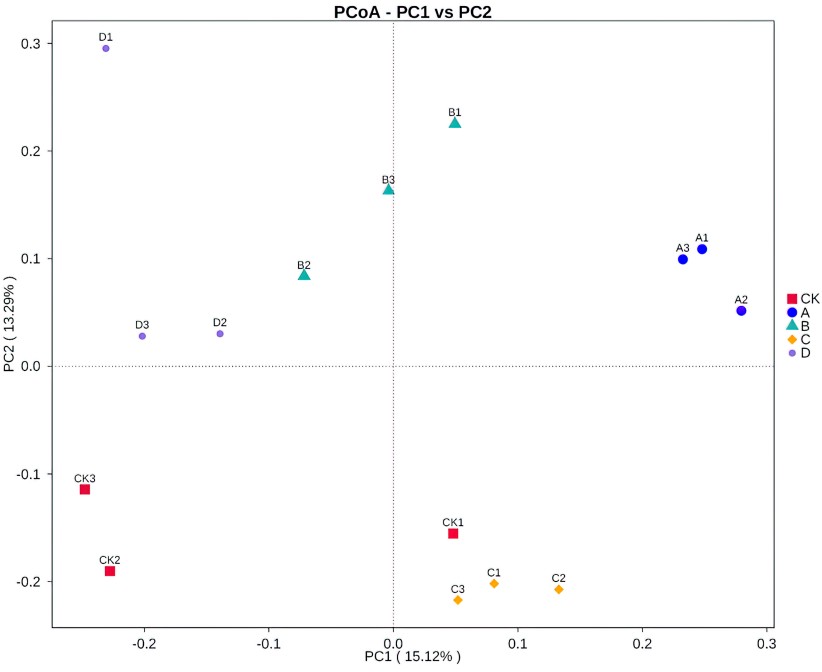

**Figure 3 Principal coordinates analysis (PCoA) based on Bray–Curtis distance method at the OTU level.** CK, A, B, C and D: bare soil, sapling, mature, overripe and deadwood, respectively; the second number representing the replicate number.

## Differences in fungal community composition at different levels of rhizosphere soil

A total of 10 phyla, 36 classes, 77 orders, 165 families, 275 genera and 353 species were identified through comparative identification of OTUs representative sequences of soil samples (Fig. 4). As shown in Fig. 4, Ascomycota was the dominant phylum in the rhizosphere soil of *P. euphratica* (average relative abundance of 58.54%), followed by Basidiomycota (15.96%). Compared with phylum classification level, the composition of rhizosphere soil fungal community in different stages differed greatly from class classification level. At the class classification level, Sordariomycetes, Pezizomycetes and Agaricomycetes were the dominant fungi, and the relative abundance of Sordariomycetes was 8.05%~34.85%,

**Table 4  Adonis test compare data samples among groups.**

| Group | F | R$^2$ | P |
|---|---|---|---|
| D-B | 1.59 | 0.284 | 0.001[**] |
| D-C | 2.298 | 0.365 | 0.1 |
| D-CK | 1.521 | 0.275 | 0.1 |
| D-A | 1.957 | 0.329 | 0.1 |
| B-C | 1.784 | 0.308 | 0.001[**] |
| B-CK | 1.17 | 0.226 | 0.3 |
| B-A | 1.228 | 0.235 | 0.101 |
| C-CK | 1.868 | 0.318 | 0.1 |
| C-A | 1.438 | 0.264 | 0.4 |
| CK-A | 1.583 | 0.264 | 0.001[**] |

**Notes.**
[**]Highly significant $p$-value, $p < 0.01$.
(CK, A, B, C and D: bare soil, sapling, mature, overripe and deadwood).

Pezizomycetes (0.21%∼36.09%) and Agaricomycetes (0.35%∼56.94%). At the order classification level, the dominant species were Pezizales, Agaricales, and Hypocreales, accounting for 0.20%∼36.08%, 0.23%∼56.84% and 1.04%∼29.02% of the total sequences in all groups, respectively. The dominant species at the family classification level are Pyronemataceae, Agaricales_fam_Incertae_sedis and unidentified_Sordariales_sp, Pyronemataceae (0.19%∼35.32%), Agaricales_fam_Incertae_sedis (0.001%∼56.43%), unidentified_Sordariales_sp (0.02%∼27.84%). The dominant species at the genus classification level were *Geopora*, *Chondrostereum* and *unidentified_Sordariales_sp*, *Geopora* (0.08%∼35.26%), *Chondrostereum* (0.001%∼56.43%), *unidentified_Sordariales_sp* (0.02%∼27.84%). The dominant species at the species classification level were *Chondrostereum_purpureum*, *Geopora_sepulta* and *Geopora_sp*, *Chondrostereum_purpureum* (0.001%∼56.43%), *Geopora_sepulta* (0.01%∼27.38%), *Geopora_sp* (0.07%∼26.52%). In conclusion, the relative expression abundance of the top ten fungi at each taxonomic level was significantly different at each stage.

## Species differences of soil fungal community

LEfSe was used to search for biomarkers, so as to find species with significant differences in abundance between groups. In this study, LEfSe analysis was used to analyze the species abundance data of fungi in rhizospheres soil samples, the rank sum test was used to detect the different species in different groups and LDA score (LDA score = 4) was obtained through LDA. Finally, the evolutionary clade of different species (Fig. 5A) and the histogram of LDA value distribution (Fig. 5B) were drawn, both of them reflected the distribution characteristics of species with different rhizosphere fungal community structure of *P. euphratica*. A total of 23 biomarkers were obtained, with relatively more in B and C samples (three taxa for A sample, eight taxa for B sample, eight taxa for C sample, and four taxa for D sample). Specifically, *Sporobolomyces sp*, *Sporobolomyces*, *Fusarium proliferatum* were significant in A sample, *Thelephoraceae sp*, Thelephorales, *unidentified*, Thelephoraceae, Sordariales, *Sordariales sp*, *unidentified Sordariales sp*, unidentified Sordariales sp were

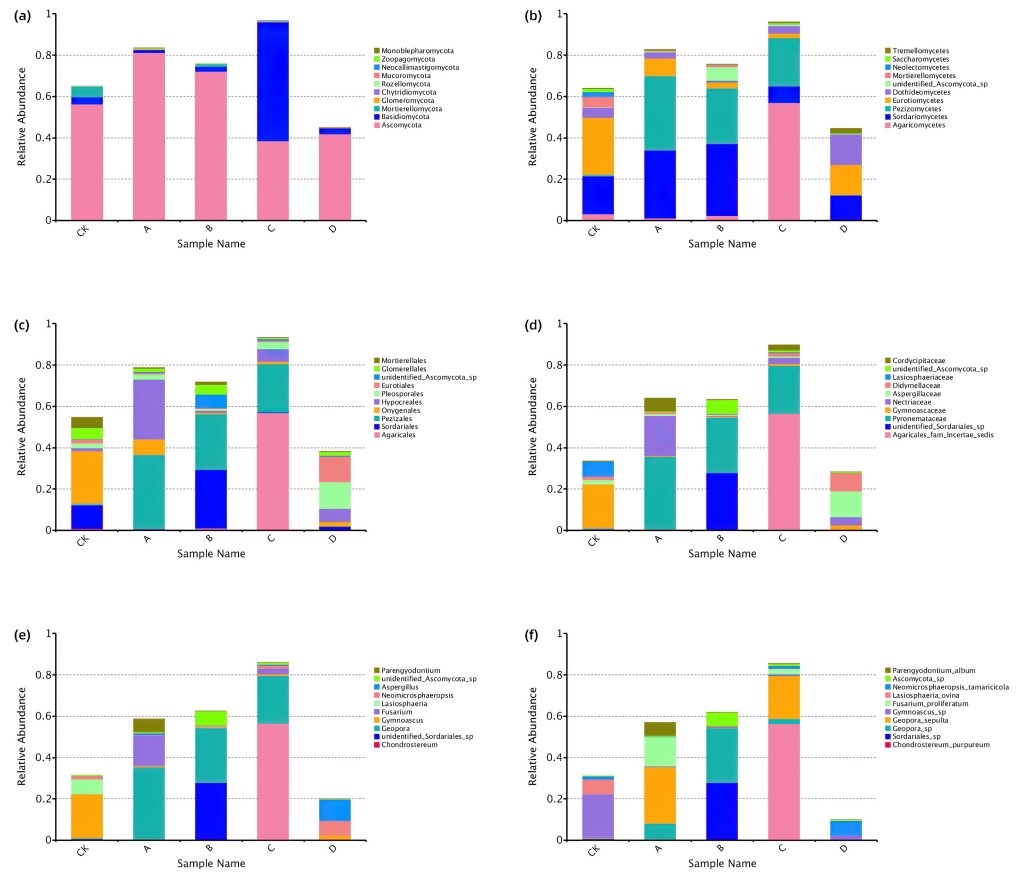

**Figure 4** **Top 10 relative abundance of fungal community classified at phylum (A), class (B), order (C), family (D), genus (E) and species (F) level in each sample.** Ordinate is the relative abundance of fungal community; abscissa is the group name (CK, A, B, C and D: bare soil, sapling, mature, overripe and dead-wood, respectively).

significant in B sample, Synchytriales, Synchytriomycetes, *Synchytrium* endobioticum, *Synchytrium*, synchytriaceae, *Chondrostereum*, *Chondrostereum purpureum*, Agaricales fam Incertae sedis were significant in C sample, Tremellomycetes, *Acremonium rutilum*, *Gibberella intricans*, *Gibberella* were significant in D sample.

## Correlation of soil physical and chemical factors and fungal community structure

Canonical correlation analysis (CCA) can reflect the relationship between microflora and environmental factors, and can obtain the important environmental driving factors that affect the distribution of samples (Fig. 6). EC, TDS and AK were the main environmental factors that significantly affected the rhizosphere fungal community of *P. euphratica* ($P$ < 0.05), and EC was the main driving factor ($R^2 = 0.704$, $P$ <0.01) (Table 5). As shown in Fig. 7, the interpretation amount of the first sorting axis was 12.15%, and that of the second sorting axis was 11.47%. Spearman's correlation analysis was used to analyze the correlation between soil factors and the relative abundance of the top 35 species at

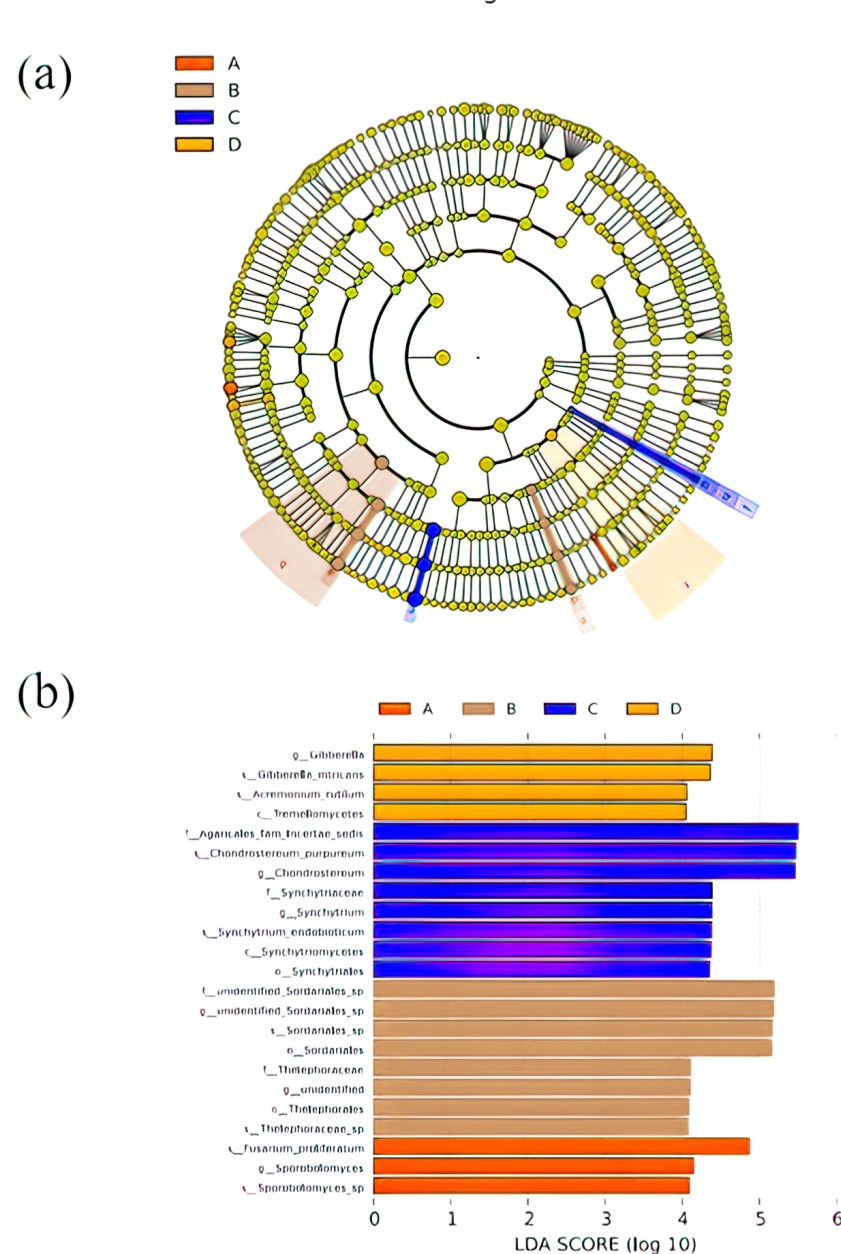

**Figure 5 Cladograms (A) and LDA value distribution histogram (B) in samples.** In cladograms (A), the circle radiating from inside to outside represents the taxonomic level from the Phylum to the species. Each small circle at a different taxonomic level represents a taxonomic at that level, and the diameter of the small circle is proportionate to the relative abundance of species. The figure shows the species with LDA Score greater than the set value (default setting is 4) (B), that is, species with significant differences in different groups. The length of the histogram represents the size of the influence of species with significant differences. The English letters in the figure is the group name (A, B, C and D: bare soil, sapling, mature, overripe and deadwood, respectively).

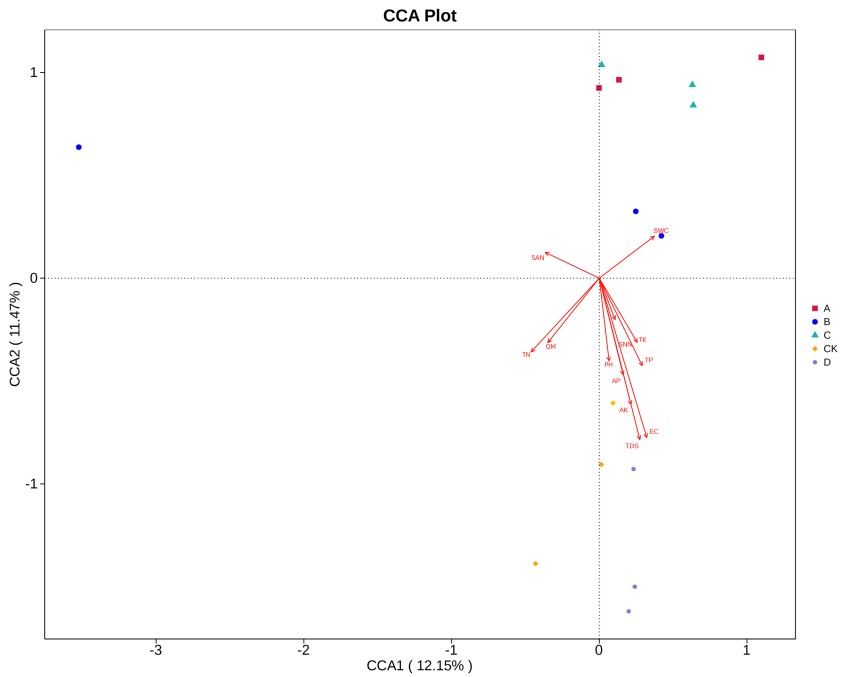

**Figure 6** **CCA ordination diagrams of fungal communities and soil variables.** The length of the arrow line represents the degree of correlation between a certain environmental factor and community and species distribution, and the longer the arrow, the greater the correlation. When the angle between the environmental factors is acute, it means that there is a positive correlation between the two environmental factors, while when the angle is obtuse, there is a negative correlation. (SWC, soil water content; OM, organic matter; TN, total nitrogen; TP, total phosphorus; TK, total potassium; AK, available potassium; AP, available phosphorus; SNN, nitrate nitrogen; SAN, ammonium nitrogen; pH, hydrogen ion concentration; EC, electrical conductivity; TDS, total salt; respectively; A, B, C, D and CK: sapling, mature, overripe, deadwood and bare soil).

genus level (Fig. 7). The results showed that EC was significantly positively correlated with *Thielavia*, *Xerombrophila* ($P <0.05$), and negatively correlated with *Lecanicillium*, *unidentified*, *unidentified Onygenales sp*, *Fusarium*, *Geopora* ($P <0.05$). TDS content was positively correlated with *Xerombrophila* ($P < 0.05$), and negatively correlated with *Lecanicillium*, *unidentified*, *unidentified Onygenales sp*, *Fusarium*, *Geopora* ($P < 0.05$). AK was positively correlated with *Thielavia*, *Xerombrophila*, *Didymella*, *Neomyrmecridium*, *Aspergillus* ($P < 0.05$), and negatively correlated with *unidentified* ($P < 0.05$).

## DISCUSSION

The lower reaches of the Tarim River is located in an extremely arid climate area, with a harsh ecological environment and a very fragile ecosystem (*Zhao et al., 2015*). Water and salt content are the key factors limiting plant growth and development in this habitat, our study found that SWC reached the highest in the root of young *P. euphratica*, followed by the deadwood, and was significantly higher than mature, overripe and bare land ($P < 0.05$), TDS content reached the highest in the rhizosphere of deadwood, which was three times of that in other growth stages, the high SWC in the rhizosphere soil of saplings helps
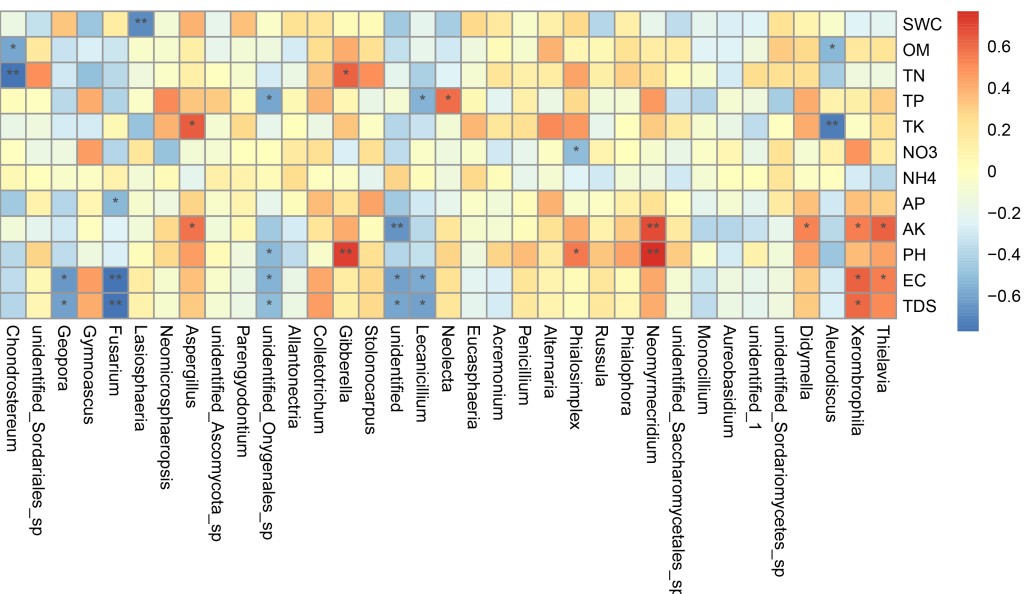

**Figure 7** **Heat maps of Spearman's correlation analysis.** Ordinate is the information of environmental factors, and abscissa is the information of species at the genera level of taxonomy. The correlation coefficient r of Spearman is between −1 and 1, $r < 0$ is negative correlation, $r > 0$ is positive correlation, and the mark * is significance test $p < 0.05$. (SWC, soil water content; OM, organic matter; TN, total nitrogen; TP, total phosphorus; TK, total potassium; AK, available potassium, AP, available phosphorus; $NO_3$, nitrate nitrogen; $NH_4$: ammonium nitrogen; pH, hydrogen ion concentration; EC, electrical conductivity; TDS, total salt).

**Table 5** **Results for CCA testing effects of soil physicochemical properties on the composition of rhizosphere fungal community in *P. euphratica*.**

|  | r² | P |
|---|---|---|
| SWC | 0.181 | 0.318 |
| OM | 0.220 | 0.202 |
| TN | 0.342 | 0.077 |
| TP | 0.266 | 0.164 |
| TK | 0.052 | 0.744 |
| SNN | 0.163 | 0.320 |
| SAN | 0.151 | 0.389 |
| AP | 0.247 | 0.180 |
| AK | 0.423 | 0.044[*] |
| PH | 0.166 | 0.371 |
| EC | 0.704 | 0.000[**] |
| TDS | 0.692 | 0.001[**] |

**Notes.**
r² reflects the relationship between soil physical and chemical properties and fungal community structure, the *P* values are the correlation coefficients.
[**] *P* < 0.01.
[*] *P* < 0.05.

them to translocate, grow and develop smoothly in this environment. With the increase of tree age and the weakening of growth potential, the capacity of plants to absorb soil salt decreased, and the salt accumulation significantly increased the salt content in the rhizosphere soil of the deadwood, it was consistent with the results of Guan's study (*Guan et al., 2020*). In addition, the accumulation of salt and some nutrients in the rhizosphere soil may be the result of the comprehensive action of roots, water, microorganisms and other factors. *P. euphratica* is a salt-tolerant plant, during its growth and development, it can selectively absorb some salt ions, especially potassium ions, so as to resist the stress of saline soil environment. Therefore, EC and AK in the rhizosphere soil of sapling, mature and overripe were significantly lower than deadwood.

In this study, Shannon index and Simpson index were used to calculate community diversity, ACE and Chao1 index were used to calculate community richness (*Bokulich et al., 2013*). Through the calculation of the above indexes, there was no significant difference between different groups. However, in terms of fungal community structure, there were significant differences between mature and overripe, mature and deadwood. According to previous studies, with the growth and development of plants, the metabolic activities of plant roots, the nutrients, water and ventilation in the environment around the roots have different changes, and the diversity of rhizosphere microorganisms and community structure also change (*Qiu et al., 2016*; *Zhao, Zhou & Ren, 2020*). Deadwood root's SWC, OM and part of the soil nutrient content values were significantly higher than other stages, the number up to 518 OTUs, alpha diversity index is relatively high, studies have shown that SWC, OM, available nutrient content is higher, can create favorable conditions for the growth of fungi, which can protect soil fungi and enhance their community abundance (*Zhang et al., 2021*).

Ascomycota and Basidiomycota were the dominant phyla, but the abundance of the two fungi was different at different developmental stages of *P. euphratica.* Among them, Ascomycota had the highest relative abundance in sapling (81.20%), followed by mature (72.08%), and the lowest in overripe (38.74%), Basidiomycota had the highest relative abundance in overripe (57.25%), and the relative abundance in other periods was low (range 1.41%~2.69%). Many studies have shown that Ascomycota and Basidiomycota were the dominant fungi in plant rhizosphere (*Chen et al., 2021*). For example, in the rhizosphere of *Picea asperata* (*Liu et al., 2021*), *Castanopsis hystrix* and *Pinus massoniana* (*Wang et al., 2021a*; *Wang et al., 2021b*), the relative abundance of both fungi were higher than other fungi. Ascomycota was the dominant fungi in soil, most of which were saprophytic fungi (*Paungfoo-Lonhienne et al., 2015*), they could degrade the organic matter such as lignin and keratin in soil (*Beimforde et al., 2014*), and have a rapid evolution rate in various soil ecosystems (*Wang & Guo, 2016*). In addition, Ascomycota might have the ability to adapt to saline alkali or relatively arid soil environment, which made it the main dominant fungal community in *P. euphratica* rhizosphere soil under the harsh environment in the lower reaches of Tarim River. Basidiomycota was mostly saprophytic or parasitic fungi, as an important decomposer in the soil (*Yelle et al., 2008*), it played an important role in the nutrient cycle of *P. euphratica* rhizosphere soil.

Compared with phylum classification level, dominant flora differed greatly among different groups at genus level. *Chondrostereum*, as a biomarker in C sample, had the highest relative abundance (56.43%) in C sample, but less than 0.01% in B and D samples. *Geopora* had the highest relative abundance in A sample (35.26%), but less than 0.1% in D sample. *Unidentified_Sordariales_sp*, as the biomarker in B sample, had the highest relative abundance in B sample (27.84%), but the average abundance in other periods is less than 0.1%. Otherwise, the abundance of the above dominant fungi in CK was less than 1%. In conclusion, dominant fungal communities changed significantly in different stages. with the growth and development of *P. euphratica*, its roots had different selective enrichment effects on specific fungi in the soil, which further indicated that these fungi may be closely related to the growth and development of *P. euphratica*. Moreover, previous studies have shown that the dominant fungal community changes dynamically with the growth and development of plants (*Li et al., 2020a*; *Li et al., 2020b*).

The microbial community structure in the rhizosphere of plants was affected by various biological and abiotic factors. The species and growth stage of plants determine which microorganisms can enrich in the rhizosphere. On the other hand, the physical and chemical properties of the soil in this area have a more macroscopic influence on the microbial community (*Chu et al., 2020*). The results showed that EC, TDS and AK had significant effects on soil fungal community, and EC was the most important factor, this is reflected in the Spearman's correlation analysis between dominant fungi and soil physical and chemical factors. EC was significantly correlated with seven dominant fungi species, of which it was significantly negatively correlated with four dominant fungi species. Some studies showed that the main soil factor affecting the metabolic characteristics of *P. euphratica* rhizosphere fungal community was EC, and EC and AK were negatively correlated with fungal metabolic activity (*Wang et al., 2017*), which were consistent with our study. Due to the scarcity of precipitation and high daily evaporation, the study area belonged to the harsh arid and salt-alkali environment. TDS not only had a certain influence on plant growth, but also could directly inhibit the activity of microorganisms (*Wang et al., 2009*), and the influence on soil fungal community should not be ignored.

## CONCLUSIONS

Based on high-throughput sequencing technology, we studied the differences in the composition and structure of soil fungal community in the rhizosphere of *P. euphratica* at four development stages. The results showed that the dominant phyla of rhizosphere fungi were Ascomycota and Basidiomycota. The dominant fungal genera were *Chondrostereum*, *unidentified_Sordariales_sp*, and *Geopora*. The relative abundance of the top 10 fungi at each classification level varied greatly in different growth stages. The study on the relationship between environmental factors and fungal community showed that EC was the main soil factor affecting the composition of rhizosphere fungal community, followed by TDS and AK.

## ACKNOWLEDGEMENTS

In this study, we would like to thank professor L.Z for her guidance, and all the authors for their joint efforts. We would like to thank Beijing Compass Biotechnology Co., Ltd. (Beijing, China) for its assistance in the analysis of the Illumina genome.

### Funding

This study was funded by the National Nature Science Foundation of China (31560177). The funders had no role in study design, data collection and analysis, decision to publish, or preparation of the manuscript.

### Grant Disclosures

The following grant information was disclosed by the authors:
National Nature Science Foundation of China: 31560177.

### Competing Interests

The authors declare there are no competing interests.

### Author Contributions

- Yuanyuan Li conceived and designed the experiments, performed the experiments, analyzed the data, prepared figures and/or tables, authored or reviewed drafts of the article, and approved the final draft.
- Hanli Dang conceived and designed the experiments, performed the experiments, analyzed the data, prepared figures and/or tables, authored or reviewed drafts of the article, and approved the final draft.
- Xinhua Lv conceived and designed the experiments, performed the experiments, prepared figures and/or tables, and approved the final draft.
- Zhongke Wang conceived and designed the experiments, performed the experiments, prepared figures and/or tables, and approved the final draft.
- Xiaozhen Pu performed the experiments, analyzed the data, authored or reviewed drafts of the article, and approved the final draft.
- Li Zhuang performed the experiments, analyzed the data, authored or reviewed drafts of the article, and approved the final draft.

### Data Availability

The data is available at NCBI SRA: PRJNA796236.

### Supplemental Information

Supplemental information for this article can be found online at http://dx.doi.org/10.7717/peerj.13552#supplemental-information.

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
