# Peer review of "High-throughput sequencing reveals rhizosphere fungal community composition and diversity at different growth stages of Populus euphratica in the lower reaches of the Tarim River"

_PeerJ, doi:10.7717/peerj.13552_

## Round 0.1 · original submission · Minor Revisions

Dear Dr. Li and co-authors,

I just received the reviews of your manuscript. Please, consider all comments and suggestions provided by both reviewers during the revision of your manuscript.

Don't forget to include a letter response along with the revised version of the manuscript. In this letter you must respond point by point to each question.

Best regards,

Xiaoming Kang

·

Basic reporting

No comment

Experimental design

No comment

Validity of the findings

Is it possible for the author to validate and justify the work carried out in this study using the following reference, specifically with reference to the use of primers and the bias in sequencing outcome?



Bellemain et al. BMC Microbiology 2010, 10:189
http://www.biomedcentral.com/1471-2180/10/189

ITS as an environmental DNA barcode for fungi: an in silico approach reveals potential PCR biases (Eva et al. 2010).

Additional comments

The research paper titled "High-throughput sequencing reveals rhizosphere fungal community composition and diversity at different growth stages of Populus euphratica in the lower reaches of the Tarim River" explains the details of the rhizosphere fungal communities at the different growth stages of P. euphraria.

The outcomes of this piece of research provide scientific basis for the detail study and importance of rhizosphere fungal communities and their interaction with plants in arid ecosystem.

This study provides a base for future investigation of differences in the composition and structure of soil fungal community in the rhizosphere of diverse ecosystem.

Reviewer 2 ·

Basic reporting

The authors are interested in the diversity, community structure, and relationship between rhizosphere fungi and environmental factors at different growth stages of Populus euphraria. The MS is basically well-written with the information clearly-presented.

Experimental design

Overall, the experimental design and data analysis seem solid.

Validity of the findings

The Result part should not take the statistical method as the sub title, please reorganized the Result part based on scientific problems mentioned above.

Additional comments

1.The Abstract needs to be rewritten, with the following parts for your reference:
Aim,
Methods,
Results, specially this part should be briefly presented important findings, not all your results.
Main conclusions

2.A figure about the location of study and samping sites is needed at the "Study sites and sampling" in M&M part.
3.Please provide higher resolution pictures in the MS.

Reviewer 3 ·

Basic reporting

The hierarchical structure of the paper “High-throughput sequencing reveals rhizosphere fungal community composition and diversity at different growth stages of Populus euphratica in the lower reaches of the Tarim River (#69932) ” is clear, and the research results have certain reference value. But there are many problems that need to be modified. After minor revision, it can be accepted for publication in "Peerj."

1.In order to clarify the difference of fungal function of spesies level between different comparison groups, if it is better to add KEGG passway t test analysis?

2. The clarity of the drawing is too poor to see the details.

3. In the part of “Soil physicochemical properties”. Please add the number of replicates for the experiment.

4. Line 36: P. euphraria ? Moreover, the abbreviate of Populus euphraria in the manuscript should be unified.

5. Line 36 have-had; is-was. In the manuscript, there were a lot of such problems in the article. Please amend.

Experimental design

no comment

Validity of the findings

no comment'

---

## Round 0.2 · Minor Revisions

Dear Dr. Li and co-authors,

Please, consider all comments and suggestions provided by both reviewers during the revision of your manuscript.

Don't forget to include a letter response along with the revised version of the manuscript. In this letter you must respond point by point to each question.

Best regards,

Xiaoming Kang

Reviewer 2 ·

Basic reporting

Overall, this version is better than before, but there are still some parts that need to be modified.
1. Abstract: Methods, is more appropriate than Methodology.
2. Results: There may be misunderstandings. My comment was “The Result part should NOT take the statistical method as the sub title”,the authors could read more papers and learn how to write a scientific paper, including the title of the results part.

Experimental design

no comment

Validity of the findings

no comment

Reviewer 3 ·

Basic reporting

The manuscript is well written. Besides the data support the conclusion and the research results have certain reference value. References are properly cited. The structure of the article and the pictures are suitable.

Experimental design

The experiment is designed rigorously. There are very interesting data in this manuscript.

Validity of the findings

The data support the conclusion and the research results have certain reference value.

---

## Round 0.3 · accepted · Accept

Dear authors,

I am pleased to inform you that, following the revision made based on the reviewer’s comments, your manuscript is now acceptable for publication in PeerJ.

Best regards

Xiaoming Kang